# Analytical Modeling of Acoustic Emission Due to an Internal Point Source in a Transversely Isotropic Cylinder

**Kwang Bok Kim, Bong Ki Kim, Sang Guk Lee**  **and Jun-Gill Kang \***

Rm #306, Integrity Diagnostics Korea (IDK), IT Venture Town, 35, Techno 9-ro, Yuseong-gu, Daejeon 34027, Korea; emitterkim@idkorea.org (K.B.K.); peterkim@idkorea.org (B.K.K.); utopialsg@idkorea.org (S.G.L.)
\* Correspondence: jgkang@cnu.ac.kr

**Abstract:** In this paper, the displacement fields responsible for acoustic emission (AE), excited from a point source in a transversely isotropic cylinder, are derived by solving the Navier-Lamé (NL) equation. The point source as an internal defect is represented by a spatiotemporal concentrated force. The introduction of three potentials correlated with the point source to displacement field vector decouples the coupled NL equation in cylindrical coordinates. Under these conditions, we solve the radial, tangential, and axial displacement fields. Analytical simulations of AE were carried out at several point source locations. Our results demonstrate that analytical modeling is a powerful tool for characterizing AE features generated from an internal defect source.

**Keywords:** acoustic emission; analytical modeling; cylinder; point source



## 1. Introduction

Acoustic emission (AE) refers to the elastic wave generated by a material as a result of a sudden release of energy (other than heat) from localized sources within a solid. Material degradation related to deformation and fracture development is classified as the primary AE source, and is usually distinguished from secondary AE sources including leak and flow, chemical reactions, and the fabrication process. Crack formation and growth are the most important AE sources in practical non-destructive testing (NDT) investigations. The formation of a new crack face is accompanied by sudden changes in stress and displacement of the material in the vicinity of the crack. Consequently, an elastic wave is generated from the tip of the crack source [1]. AE testing has several unique advantages over other NDT methods, the most significant of which is its real-time monitoring capability; this allows for the detection of damage and degradation in various materials and structures [2–6]. AE data, refined with appropriate algorithms [7–11], provide useful information about the source of the emission and progression of crack growth; however, a quantitative interpretation of the AE signal is required to understand the physical processes underlying AE features. Analytical modeling of the AE signal is very important; however, in cylindrical structures, it is difficult to predict the AE generated from an internal crack.

The point source, defined as a body force *f* acting at a point, has been adapted as an AE excitation source in seismic displacements, crack formation and fracture, and concentrated vertical step force [12–16]. In the elastic field, the point source has been treated as a concentrated time harmonic source [17]. Although the AE generated by the point source is important for characterizing real signals observed in practical NDT investigations, theoretical modeling has been limited to spherical geometries with an infinite domain, given that a homogeneous solution is obtainable from the three-dimensional wave equation in spherical coordinates [12,13,17]. Most theoretical works on elastic wave propagation in cylindrical coordinates have focused on situations with or without external perturbations [18–22]. In cylindrical structures, the displacement field has been formulated in terms of one compression (P) potential, and two shear (vertically polarized, SV, and horizontally polarized, SH)

potentials, by using the models proposed by Morse and Feshbach [23] and Buchwald [24], respectively. The basic difference between the two models is that the compression and shear parts are separated in one model, but not the other. The two models have been examined comprehensively by Honarvar et al. [18,19] and Sakhr et al. [21,22]. In contrast to cases that are unperturbed or perturbed by external forces, the three potentials should be incorporated along with the concentrated force (CF) exerted by the internal defects [17,25].

In this study, a "concentrated force-incorporated potential" (CFIP) is introduced into the displacement field on the basis of the model proposed by Morse and Feshbach [23], and the Navier-Lamé (NL) equation is constructed (which involves the CF in a transversely isotropic cylinder [TIC]). The CFIP allows the NL equation to be separated into three partial differential equations (PDEs), representing the potentials for P, SV, and SH waves. As an internal defect, the CF has both spatial and temporal properties, represented by the delta function and harmonic oscillation of the point source, respectively. For solving the NL equation, determining Green's function for the delta function in cylindrical coordinates is the first task. Mohammad et al. proposed a Green's function for a closed cylinder, using "the method of separation of variables" [26]. We reconstructed Green's function for the delta function by applying the continuity and discontinuity principles to the boundary around the point source. Complete formation of the CF allowed us to solve the three potentials generated by the point source in the closed cylinder. Absolute values of the displacement fields were calculated by applying a fundamental set of boundary conditions in the cylinder to the solutions, in which two modes of CF along the radial and axial directions were considered. To our knowledge, no theoretical work on AE, excited by an internal point source in cylindrical geometries, has been presented in the literature. This paper establishes a mathematical model that provides insight into the overall process of the AE signal from generation and propagation to reception. These studies can be employed for evaluating the AE signal generated from an internal crack in a cylindrical structure.

## 2. Green's Function

The body force arising from the point source located at $x_0$ is mathematically formulated in terms of an oscillating impulse with natural frequencies of the material in a given geometry, as follows:

$$f = P(t)\delta(x - x_0)e^{-i\omega t}, \tag{1}$$

where $P(t)$ is the CF vector, $\delta(x - x_0)$ is the delta function, and $\omega$ is the predominant angular frequency ($\omega = 2\pi\nu$) of AE. Green's function $g(x; x_0)$, as the solution of the delta function, is defined as

$$\nabla^2 g(x; x_0) = \delta(x - x_0). \tag{2}$$

The force can be rewritten as

$$f = \nabla^2 P g(x; x_0) = \nabla[\nabla \cdot P g(x; x_0)] - \nabla \times [\nabla \times P g(x; x_0)]. \tag{3}$$

In cylindrical coordinates, Equation (2) is expressed as

$$\nabla^2 g(r, \theta, z; r_0, \theta_0, z_0) = \frac{\delta(r - r_0)\delta(\theta - \theta_0)\delta(z - z_0)}{r}. \tag{4}$$

If $r - r_0 \neq 0, \theta - \theta_0 \neq 0$, and $z - z_0 \neq 0$,

$$\nabla^2 g(x; x_0) = 0. \tag{5}$$

Green's function is separable in a cylindrical coordinate system as

$$g(x; x_0) = g_r(r; r_0)g_\theta(\theta; \theta_0)g_z(z; z_0). \tag{6}$$

Substituting the Laplacian in cylindrical coordinates into Equation (5) gives the following:

$$\frac{r^2}{g_r}\frac{\partial^2 g_r}{\partial r^2} + \frac{r}{g_r}\frac{\partial g_r}{\partial r} + \frac{1}{g_\theta}\frac{\partial^2 g_\theta}{\partial \theta^2} + \frac{r^2}{g_z}\frac{\partial^2 g_z}{\partial z^2} = 0. \tag{7}$$

Letting

$$\frac{1}{g_z}\frac{\partial^2 g_z}{\partial z^2} = \kappa_z^2, \tag{8}$$

$$\frac{1}{g_\theta}\frac{\partial^2 g_\theta}{\partial \theta^2} = -v^2, \tag{9}$$

then Equation (7) has the form

$$r^2\frac{\partial^2 g_r}{\partial r^2} + r\frac{\partial g_r}{\partial r} + \left(\kappa_z^2 r^2 - v^2\right)g_r = 0. \tag{10}$$

The solution of Equation (8) can be formulated in two regions:

$$g_z(z;z_0) = \begin{cases} A_z(z_0)e^{\kappa_z z} & 0 < z < z_0 < l \\ B_z(z_0)e^{-\kappa_z z} & 0 < z_0 < z < l \end{cases} \tag{11}$$

The continuity and discontinuity principles are applied to find coefficients $A_z$ and $B_z$. The continuity principle states that Green's function is continuous around the point source, i.e.,

$$g_z(z;z_0)|_{z=z_0^+} = g_z(z;z_0)|_{z=z_0^-}. \tag{12}$$

Equation (12) leads to

$$A_z(z_0)e^{\kappa_z z_0} = B_z(z_0)e^{-\kappa_z z_0}. \tag{13}$$

Although Green's function is discontinuous at the point source, the difference in Green's function between the adjacent two points, $z_0 \pm \varepsilon$, around the point source is unity. When $\varepsilon$ becomes 0, the discontinuity principle leads to

$$\left.\frac{\partial g_z(z;z_0)}{\partial z}\right|_{z=z_0^+} - \left.\frac{\partial g_z(z;z_0)}{\partial z}\right|_{z=z_0^-} = 1. \tag{14}$$

As a result, we obtain

$$-\kappa_z B_z(z_0)e^{-\kappa_z z_0} - \kappa_z A_z(z_0)e^{\kappa_z z_0} = 1. \tag{15}$$

From Equations (13) and (15),

$$A_z(z_0) = -\frac{1}{2\kappa_z}e^{-\kappa_z z_0} \quad \text{and} \quad B_z(z_0) = -\frac{1}{2\kappa_z}e^{\kappa_z z_0}.$$

Equation (11) becomes

$$g_z(z;z_0) = \begin{cases} -\frac{1}{2\kappa_z}e^{-\kappa_z(z_0-z)} & 0 < z < z_0 < l \\ -\frac{1}{2\kappa_z}e^{-\kappa_z(z-z_0)} & 0 < z_0 < z < l \end{cases}. \tag{16}$$

The solution of Equation (9) can be written as

$$g_\theta(\theta;\theta_0) = \begin{cases} A_\theta(\theta_0)\cos v\theta + B_\theta(\theta_0)\ \sin v\theta & 0 < \theta < \theta_0 < 2\pi \\ C_\theta(\theta_0)\cos v\theta + D_\theta(\theta_0)\ \sin v\theta & 0 < \theta_0 < \theta < 2\pi \end{cases}. \tag{17}$$

Applying the continuity principle to Equation (17) results in

$$A_\theta(\theta_0) = C_\theta(\theta_0) \quad \text{and} \quad B_\theta(\theta_0) = D_\theta(\theta_0). \tag{18}$$

The second restriction for $g_\theta(\theta; \theta_0)$ is the symmetry between the point source and observation point, i.e.,

$$g_\theta(\theta_0; \theta) = g_\theta(\theta; \theta_0). \tag{19}$$

Equation (19) leads to

$$A_\theta(\theta) \cos v\theta_0 + B_\theta(\theta) \sin v\theta_0 = A_\theta(\theta_0) \cos v\theta + B_\theta(\theta_0) \sin v\theta.$$

To satisfy the conservation law,

$$A_\theta(\theta_0) = \cos v\theta_0 \text{ and } B_\theta(\theta_0) = \sin v\theta_0. \tag{20}$$

Equations (18) and (20) result in

$$g_\theta(\theta; \theta_0) = [\cos v\theta_0 \cos v\theta + \sin v\theta_0 \sin v\theta] = \cos[v(\theta - \theta_0)].$$

For a complete cylinder, $g_\theta(\theta; \theta_0)$ is periodic, with a period of $2\pi$:

$$\cos[v(2\pi + \theta - \theta_0)] = \cos[v(\theta - \theta_0)].$$

Therefore,

$$v = 0, \pm 1, \pm 2, \cdots.$$

The final form of Equation (17) is

$$g_\theta(\theta; \theta_0) = \cos[v(\theta - \theta_0)] \quad (v = 0, \pm 1, \pm 2, \cdots). \tag{21}$$

The solution of Equation (10) is a typical Bessel function, given by

$$g_r(r; r_0) = \begin{cases} A_{vr}(r_0) J_v(\kappa_z r) & 0 < r < r_0 < a \\ B_{vr}(r_0) J_v(\kappa_z r) & 0 < r_0 < r < a \end{cases} \tag{22}$$

where $J_v$ is the first kind of Bessel function of $v$-th order. Note that the second kind of Bessel function is excluded because it has a singularity at the origin, which is included in the domain of the cylinder. Applying the continuity principle to Equation (22), i.e., $g_r(r; r_0)|_{r=r_0^+} = g_r(r; r_0)|_{r=r_0^-}$, results in

$$A_{vr}(r_0) J_v(\kappa_z r_0) = B_{vr}(r_0) J_v(\kappa_z r_0).$$

We find

$$A_{vr}(r_0) = B_{vr}(r_0), \tag{23}$$

where $r_0 = \sqrt{x_{0i}^2 + x_{0j}^2}$. In addition, applying the symmetry principle to Equation (22) gives $g_r(r_0; r) = g_r(r; r_0)$, with

$$A_{vr}(r) J_v(\kappa_z r_0) = A_{vr}(r_0) J_v(\kappa_z r).$$

we obtain

$$A_{vr}(r_0) = J_v(\kappa_z r_0). \tag{24}$$

Substituting Equations (23) and (24) into Equation (22) gives

$$g_r(r; r_0) = J_v(\kappa_z r_0) J_v(\kappa_z r) \quad (0 < r < a). \tag{25}$$

Applying the boundary condition, i.e.,

$$g_r(r; r_0)|_{r=a-r_0} = 0, \qquad g_r(r; r_0)|_{r=r_0} \neq 0,$$

to Equation (25) gives $J_v[\kappa_z(a - r_0)] = 0$. Denoting the $n$-th root of the first kind of Bessel function of the $v$-th order as $r_{vn}$, i.e., $J_v(r_{vn}) = 0$, then $r_{vn} = \kappa_z(a - r_0)$ $(n = 1, 2, \cdots)$. Since the first root of the first kind of Bessel function is most significant, we select $n = 1$.

$$\kappa_z = \frac{r_{v1}}{a - r_0}. \tag{26}$$

Introducing a parameter, $A_{v1}$, to Equation (25) gives

$$g_r(r; r_0) = A_{v1} J_v\left(\frac{r_{v1}}{a - r_0} r_0\right) J_v\left(\frac{r_{v1}}{a - r_0} r\right) \quad (0 < r < a). \tag{27}$$

From Equations (16), (21) and (27), the formula of the Green's function for the Kronecker delta function becomes

$$\begin{aligned}
g(r, \theta, z; r_0, \theta_0, z_0) &= A_{v1} J_v\left(\frac{r_{v1}}{a-r_0} r_0\right) J_v\left(\frac{r_{v1}}{a-r_0} r\right) \cos[v(\theta - \theta_0)] \\
&\times \begin{cases} -\frac{a-r_0}{2r_{v1}} e^{-\frac{r_{v1}}{a-r_0}(z_0 - z)} & 0 < z < z_0 < l \\ -\frac{a-r_0}{2r_{v1}} e^{-\frac{r_{v1}}{a-r_0}(z - z_0)} & 0 < z_0 < z < l \end{cases}.
\end{aligned} \tag{28}$$

The next task is to find the constant $A_{v1}$ in Equation (28). Equation (4) is rewritten as

$$\begin{aligned}
&\nabla^2 g(r, \theta, z; r_0, \theta_0, z_0) \\
&= \nabla^2 g_r(r; r_0) g_\theta(\theta; \theta_0) g_z(z; z_0) \\
&= \frac{g_\theta g_z}{r} \frac{\partial}{\partial r}\left(r \frac{\partial g_r}{\partial r}\right) + \frac{g_r g_z}{r^2} \frac{\partial^2 g_\theta}{\partial \theta^2} + g_r g_\theta \frac{\partial^2 g_z}{\partial z^2} \\
&= \frac{\delta(r - r_0)\delta(\theta - \theta_0)\delta(z - z_0)}{r}.
\end{aligned} \tag{29}$$

First, solve $\delta(z - z_0)$ by integrating both sides over $(0, l)$

$$\begin{aligned}
&\int_{z=0}^{z=l} \left[\frac{g_\theta g_z}{r} \frac{\partial}{\partial r}\left(r \frac{\partial g_r}{\partial r}\right) + \frac{g_r g_z}{r^2} \frac{\partial^2 g_\theta}{\partial \theta^2} + g_r g_\theta \frac{\partial^2 g_z}{\partial z^2}\right] dz \\
&= g_r g_\theta \int_{z=0}^{z=l} \frac{\partial^2 g_z}{\partial z^2} dz.
\end{aligned}$$

Therefore,

$$g_r g_\theta \left.\frac{\partial g_z}{\partial z}\right|_0^{z=l} = \frac{\delta(r - r_0)\delta(\theta - \theta_0)}{r}. \tag{30}$$

From Equations (16), (26), and (30), we obtain

$$\begin{aligned}
g_r g_\theta \left.\frac{\partial g_z}{\partial z}\right|_0^{z=l} &= g_r g_\theta \left[-\frac{1}{2} e^{-\frac{r_{v1}}{a-r_0}(z_0 - z)}\Big|_0^{z_0} + \frac{1}{2} e^{-\frac{r_{v1}}{a-r_0}(z - z_0)}\Big|_{z_0}^{l}\right] \\
&= \frac{g_r g_\theta}{2}\left[e^{-\frac{r_{v1}}{a-r_0} z_0} + e^{-\frac{r_{v1}}{a-r_0}(l - z_0)} - 2\right],
\end{aligned}$$

and

$$\frac{g_r g_\theta}{2}\left[e^{-\frac{r_{v1}}{a-r_0} z_0} + e^{-\frac{r_{v1}}{a-r_0}(l - z_0)} - 2\right] = \frac{\delta(r - r_0)\delta(\theta - \theta_0)}{r}. \tag{31}$$

Substituting Equation (21) into the above equation and integrating both sides over $(0, 2\pi)$ gives

$$\begin{aligned}
&\frac{g_r}{2}\left[e^{-\frac{r_{v1}}{a-r_0} z_0} + e^{-\frac{r_{v1}}{a-r_0}(l - z_0)} - 2\right] \sum_{v=-\infty}^{\infty} \int_0^{2\pi} \cos[v(\theta - \theta_0)] d\theta \\
&= \frac{\delta(r - r_0)}{r} \int_0^{2\pi} \delta(\theta - \theta_0) d\theta.
\end{aligned}$$

Since

$$\int_0^{2\pi} \cos[v(\theta - \theta_0)] d\theta = \begin{cases} 2\pi & \text{if } \theta = \theta_0 \\ 0 & \text{otherwise} \end{cases},$$

the above equation is given by

$$\pi g_r \left[ e^{-\frac{r_{v1}}{a-r_0} z_0} + e^{-\frac{r_{v1}}{a-r_0}(l-z_0)} - 2 \right] = \frac{\delta(r-r_0)}{r}. \tag{32}$$

Substituting Equation (27) into Equation (32) results in

$$\pi \left[ e^{-\frac{r_{v1}}{a-r_0} z_0} + e^{-\frac{r_{v1}}{a-r_0}(l-z_0)} - 2 \right] A_{v1} \, J_v \left( \frac{r_{v1}}{a-r_0} r_0 \right) J_v \left( \frac{r_{v1}}{a-r_0} r \right) = \frac{\delta(r-r_0)}{r}. \tag{33}$$

Multiplying both sides of Equation (33) by $r J_p \left( \frac{r_{v1}}{a-r_0} r \right)$ and integrating over $(0, a - r_0)$ gives

$$\pi \left[ e^{-\frac{r_{v1}}{a-r_0} z_0} + e^{-\frac{r_{v1}}{a-r_0}(l-z_0)} - 2 \right]$$
$$\times A_{v1} \, J_v \left( \frac{r_{v1}}{a-r_0} r_0 \right) \int_0^{a-r_0} J_v \left( \frac{r_{v1}}{a-r_0} r \right) J_p \left( \frac{r_{v1}}{a-r_0} r \right) r dr \tag{34}$$
$$= \int_0^a \delta(r-r_0) J_p \left( \frac{r_{v1}}{a-r_0} r \right) d.$$

Given [26]

$$\int_0^{a-r_0} J_\mu \left( \frac{\alpha_\mu}{a-r_0} r \right) J_v \left( \frac{\alpha_v}{a-r_0} r \right) r dr = \frac{(a-r_0)^2}{2} \left[ J_{\mu+1}(\alpha_\mu) \right]^2 \delta_{\mu v}$$

and

$$\int_0^{a-r_0} \delta(r-r_0) J_\mu(r) dr = J_\mu(r_0),$$

Equation (34) can be rewritten as

$$\pi \left[ e^{-r_{v1} z_0/(a-r_0)} + e^{-r_{v1}(l-z_0)/(a-r_0)} - 2 \right] A_{v1} \, J_v \left( \frac{r_{v1}}{a-r_0} r_0 \right) \frac{(a-r_0)^2}{2} \left[ J_{v+1}(r_{v1}) \right]^2$$
$$= J_v \left( \frac{r_{v1}}{a-r_0} r_0 \right). \tag{35}$$

From Equation (35), we obtain the constant

$$A_{v1} = \frac{2}{\pi(a-r_0)^2 \left[ e^{-r_{v1} z_0/(a-r_0)} + e^{-r_{v1}(l-z_0)/(a-r_0)} - 2 \right] \left[ J_{v+1}(r_{v1}) \right]^2}. \tag{36}$$

Finally, we obtain the Green's function for the Kronecker delta function

$$g(\boldsymbol{r}, \boldsymbol{\theta}, \boldsymbol{z}; \boldsymbol{r}_0, \boldsymbol{\theta}_0, \boldsymbol{z}_0) = G_{v1} \, J_v \left( \frac{r_{v1}}{a-r_0} r_0 \right) J_v \left( \frac{r_{v1}}{a-r_0} r \right) \{ \cos[v(\theta - \theta_0)] \}$$
$$\times \begin{cases} e^{-\frac{r_{v1}(z_0-z)}{a-r_0}} & (0 < z < z_0 < l) \\ e^{-\frac{r_{v1}(z-z_0)}{a-r_0}} & (0 < z_0 < z < l) \end{cases}, \tag{37}$$

where

$$G_{v1} = -\frac{a-r_0}{2r_{v1}} A_{v1} = -\frac{1}{\pi(a-r_0) r_{v1} \left[ e^{-r_{v1} z_0/(a-r_0)} + e^{-r_{v1}(l-z_0)/(a-r_0)} - 2 \right] \left[ J_{v+1}(r_{v1}) \right]^2}. \tag{38}$$

For an azimuthally independent force ($v = 0$), Equation (37) becomes

$$g(\boldsymbol{r}, \boldsymbol{z}; \boldsymbol{r}_0, \boldsymbol{z}_0) = G_1 \, J_0 \left( \frac{r_{01}}{a-r_0} r \right), \tag{39}$$

where

$$G_1 = -\frac{J_0\left(\frac{r_{01}}{a-r_0}r_0\right)}{\pi(a-r_0)r_{01}\left[e^{-r_{v1}z_0/(a-r_0)}+e^{-r_{v1}(l-z_0)/(a-r_0)}-2\right][J_1(r_{01})]^2}$$
$$\times \begin{cases} e^{-\frac{r_{01}(z_0-z)}{a-r_0}} & (0 < z < z_0 < l) \\ e^{-\frac{r_{01}(z-z_0)}{a-r_0}} & (0 < z_0 < z < l) \end{cases}. \tag{40}$$

For convenience, we introduce new coordinates, defined as $\xi = r - r_0$, $\vartheta = \theta - \theta_0$ and $\eta = z - z_0$, where the location of the point source is the new origin at $\xi_0 = 0$, $\vartheta_0 = 0$ and $\eta_0 = 0$. Thus, Equations (39) and (40) are rewritten as

$$g(\xi, \vartheta, \eta) = G_1 J_0\left(\frac{r_{01}}{a-r_0}\xi\right), \tag{41}$$

$$G_1 = -\frac{J_0\left(\frac{r_{01}}{a-r_0}\xi_0\right)}{\pi(a-r_0)r_{01}\left[e^{-r_{v1}z_0/(a-r_0)}+e^{-r_{v1}(l-z_0)/(a-r_0)}-2\right][J_1(r_{01})]^2}$$
$$\times \begin{cases} e^{\frac{r_{01}\eta}{a-r_0}}\left( z_0 - l < \eta < 0\right) \\ e^{-\frac{r_{01}\eta}{a-r_0}}\left(0 < \eta < l - z_0\right) \end{cases}. \tag{42}$$

## 3. Displacement Fields Generated by a Point Source

The NL equation governing a wave in an elastic and homogeneous medium subject to a local body force $f$ can be written in vector form, as follows [27]

$$(\lambda + 2\mu)\nabla(\nabla \cdot u) - \mu\nabla \times (\nabla \times u) + f = \rho\frac{\partial^2 u}{\partial t^2}, \tag{43}$$

where $u$ is the displacement vector, $\lambda$ and $\mu$ are the Lamé constants, and $\rho$ is the density of the media. In this paper, the displacement field $u$ in cylindrical coordinates proposed by Morse and Feshbach [23] is given as

$$u = \nabla\Phi + \nabla \times (X\hat{e}_z) + a\nabla \times \nabla \times (\Psi\hat{e}_z), \tag{44}$$

where $\Phi$ is a scalar potential for the compressional wave (P), $X\hat{e}_z$ is a vector potential for the SH wave, $\Psi\hat{e}_z$ is a vector potential for the SV wave, and $a$ is the radius of the cylinder. When the displacement field is generated by an intrinsic point defect, the three potentials are correlated with the force vector $P$ (referred to as CFIP) as

$$\Phi = \nabla \cdot P\phi, \tag{45}$$

$$X\hat{e}_z = -\nabla \times P\chi, \tag{46}$$

$$\Psi\hat{e}_z = \nabla \times P\psi, \tag{47}$$

where $\phi$, $\chi$ and $\psi$ are unknown scalar functions, and the negative sign for the SH potential is used for further applications. Factoring the spatial and temporal parts in the potentials leads to

$$\phi(x, t; x_0) = \phi(x; x_0)e^{-i\omega t}, \tag{48}$$

$$\chi(x, t; x_0) = \chi(x; x_0)e^{-i\omega t}, \tag{49}$$

$$\psi(x, t; x_0) = \psi(x; x_0)e^{-i\omega t}, \tag{50}$$

The displacement vector can be written as:

$$u(x, t; x_0, t_0) = u(r, \theta, z, t; r_0, \theta_0, z_0, t_0)$$
$$= [\nabla(\nabla \cdot P\phi) - \nabla \times (\nabla \times P\chi) - a\nabla \times \nabla \times (\nabla \times P\psi)]e^{-i\omega t}, \tag{51}$$

where $\phi = \phi(x; x_0)$, $\chi = \chi(x; x_0)$ and $\psi = \psi(x; x_0)$. By substituting Equation (51) into Equation (43) and applying some manipulation, such as $\boldsymbol{\nabla} \cdot (\boldsymbol{\nabla}\phi) = \nabla^2\phi$, $\boldsymbol{\nabla} \times \phi = 0$, $\boldsymbol{\nabla} \cdot \psi = \boldsymbol{\nabla} \cdot \chi = 0$, and $\boldsymbol{\nabla} \times (\boldsymbol{\nabla} \cdot \boldsymbol{\nabla}\psi) = \boldsymbol{\nabla} \times (\boldsymbol{\nabla} \cdot \boldsymbol{\nabla}\chi) = 0$, Equation (43) can be rewritten as

$$\begin{aligned} \boldsymbol{\nabla}\boldsymbol{\nabla} \cdot &\left[ (\lambda + 2\mu)\left(\nabla^2 P\phi\right) + \rho\omega^2(P\phi) + Pg \right] \\ &+ a\boldsymbol{\nabla} \times \left[ \mu\nabla^4(P\psi) + \rho\omega^2\nabla^2(P\psi) \right] \\ &+ \boldsymbol{\nabla} \times \boldsymbol{\nabla} \times \left[ \mu\nabla^2(P\chi) + \rho\omega^2(P\chi) + Pg \right] = 0. \end{aligned} \tag{52}$$

In the above equation, the components in the gradient, the curl, and the curl of curl, can be independently zero

$$(\lambda + 2\mu)\nabla^2\phi + \rho\omega^2\phi + g = 0, \tag{53}$$

$$\mu\nabla^4\psi + \rho\omega^2\nabla^2\psi = 0, \tag{54}$$

$$\mu\nabla^2\chi + \rho\omega^2\chi + g = 0. \tag{55}$$

Introducing the longitudinal wave (P) speed $\left(c_P = \sqrt{\lambda + 2\mu/\rho}\right)$ and transverse wave (S) speed $\left(c_S = \sqrt{\mu/\rho}\right)$ into the above equations leads to

$$\nabla^2\phi + k_p^2\phi + \frac{k_p^2}{\rho\omega^2}g = 0, \tag{56}$$

$$\nabla^2\psi + k_s^2\psi = 0, \tag{57}$$

$$\nabla^2\chi + k_s^2\chi + \frac{k_s^2}{\rho\omega^2}g = 0, \tag{58}$$

where $k_p = \frac{\omega}{c_P}$ and $k_s = \frac{\omega}{c_S}$, correspond to the angular wavenumbers of the P and the S waves, respectively. Taking into account the Laplacian operator in the $(\xi, \vartheta, \eta)$ cylindrical coordinates, the three scalar spatial potentials have the following forms:

$$\phi(\xi, \vartheta, \eta) = \phi_r(\xi)\phi_\theta(\vartheta)\phi_z(\eta), \tag{59}$$

$$\chi(\xi, \vartheta, \eta) = \chi_r(\xi)\chi_\theta(\vartheta)\chi_z(\eta), \tag{60}$$

$$\psi(\xi, \vartheta, \eta) = \psi_r(\xi)\psi_\theta(\vartheta)\psi_z(\eta). \tag{61}$$

First, let us solve PDE of Equation (56). Using a similar method to that applied to the case of Green's function, the axial and the angular parts can be obtained as

$$\phi_z(\eta) = A_z e^{-ik_\eta\eta}, \tag{62}$$

$$\phi_\theta(\vartheta) = A_{m\theta}\cos m\vartheta + B_{m\theta}\sin m\vartheta \quad (m = 0, \pm 1, \pm 2, \cdots), \tag{63}$$

respectively. Substituting the Laplacian operator and Equations (41), (59), (62), and (63), and introducing some straightforward algebra to Equation (56) leads to

$$\frac{\partial^2\phi_r}{\partial\xi^2} + \frac{1}{\xi}\frac{\partial\phi_r}{\partial\xi} + \left(\alpha^2 - \frac{m^2}{\xi^2}\right)\phi_r = -\frac{k_p^2}{\rho\omega^2}\left(\frac{1}{\phi_\vartheta\phi_\eta}\right)G_1 J_0\left(\frac{r_{01}}{a - r_0}\xi\right), \tag{64}$$

where $\alpha^2 = k_p^2 - k_\eta^2$. Equation (64) is a second-order linear nonhomogeneous PDE, the solution of which is a linear combination of the homogeneous $(\phi_{rh})$ and particular $(\phi_{rp})$ solutions. The homogeneous solution of Equation (64) is

$$\phi_{rh} = \left\{ A_{mr}\begin{bmatrix} J_m(\alpha\xi) \\ I_m(\alpha\xi) \end{bmatrix} + B_{mr}\begin{bmatrix} Y_m(\alpha\xi) \\ K_m(\alpha\xi) \end{bmatrix} \right\} \quad \begin{array}{l} (\alpha^2 \geq 0) \\ (\alpha^2 < 0) \end{array},$$

where $J_m$ and $I_m$ are the first kind and its modified Bessel functions, respectively, and $Y_m$ and $K_m$ are the second kind and its modified Bessel functions, respectively. The Bessel functions of the second kind are excluded due to the singularity at the origin included in the domain of the cylinder. For the case of $\alpha^2 \geq 0$,

$$\phi_{rh} = A_{mr} J_m(\alpha \xi). \tag{65}$$

According to Korenev's solutions [28], the particular solution of the inhomogeneous equation has the form

$$\phi_{rp} = -\frac{k_p^2}{\rho\omega^2}\left(\frac{1}{\phi_\vartheta \phi_\eta}\right) G_1 \frac{1}{1 - \left[\frac{r_{01}}{\alpha(a-r_0)}\right]^2} J_m\left[\frac{r_{01}}{\alpha(a-r_0)}\xi\right], \qquad \frac{r_{01}}{\alpha(a-r_0)} \neq 1. \tag{66}$$

The linear combination of Equations (65) and (66) results in

$$\phi(\xi, \vartheta, \eta) = \phi_r(\xi)\phi_\theta(\vartheta)\phi_z(\eta)$$
$$= [A_{mr} J_m(\alpha \xi)][A_{m\theta}\cos(m\vartheta)]\left(A_z e^{-ik_\eta \eta}\right) - \frac{k_p^2}{\rho\omega^2} G_1 \frac{1}{1 - \left[\frac{r_{01}}{\alpha(a-r_0)}\right]^2} J_m\left[\frac{r_{01}}{\alpha(a-r_0)}\xi\right].$$

For further applications, the above equation is rewritten as

$$\phi(\xi, \vartheta, \eta) = A_m\, J_m(\alpha \xi)\cos(m\vartheta)e^{-ik_\eta \eta} - \frac{k_p^2}{\rho\omega^2} G_1 \frac{1}{1 - \left[\frac{r_{01}}{\alpha(a-r_0)}\right]^2} J_m\left[\frac{r_{01}}{\alpha(a-r_0)}\xi\right], \tag{67}$$

where the coupling constant $A_m = A_{mr}A_{m\theta}A_z$.

Following a similar procedure, the functions of $\chi(\xi, \vartheta, \eta)$ and $\psi(\xi, \vartheta, \eta)$ can be obtained as

$$\chi(\xi, \vartheta, \eta) = B_m J_m(\beta \xi)\sin(m\vartheta)e^{-ik_\eta \eta} - \frac{k_p^2}{\rho\omega^2} G_1 \frac{1}{1 - \left[\frac{r_{01}}{\beta(a-r_0)}\right]^2} J_m\left[\frac{r_{01}}{\beta(a-r_0)}\xi\right], \tag{68}$$

$$\psi(\xi, \vartheta, \eta) = C_m\, J_m(\beta \xi)\cos(m\vartheta)e^{-ik_z \eta}. \tag{69}$$

As shown in Figure 1, we introduce force vector $\boldsymbol{P}$ acting in the radial and the axial directions to solve for $\Phi$. Equation (45) can be rewritten as

$$\Phi = \boldsymbol{\nabla}\cdot\boldsymbol{P}\phi = \frac{\partial(\boldsymbol{P}\phi)}{\partial \xi_i} + \frac{\partial(\boldsymbol{P}\phi)}{\partial \xi_j} + \frac{\partial(\boldsymbol{P}\phi)}{\partial \eta}, \tag{70}$$

where $\xi_i = x_i - x_{0i}$, $\xi_j = x_j - x_{0j}$, and $\eta = x_z - x_{0z}$.

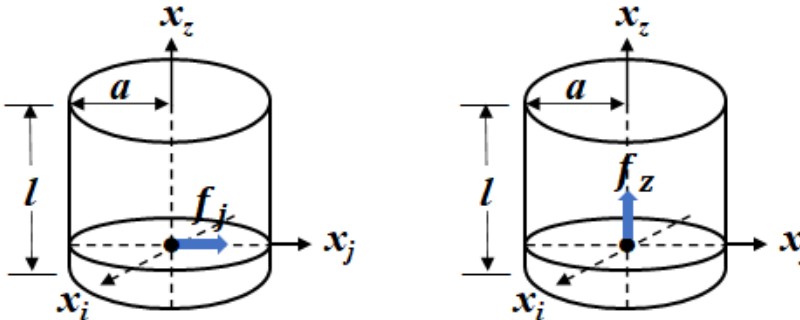

**Figure 1.** Two forms of the point source vector along the $x_j$ and $x_z$ directions used in analytical modeling.

For $\boldsymbol{P} = P_j$,

$$\Phi_j = P_j \Xi \frac{\partial \phi}{\partial \xi}, \tag{71}$$

where

$$\Xi = \frac{\xi_i}{\xi} \delta_{ij} + \frac{\xi_j}{\xi} = \frac{a \cos\varphi - x_{0i}}{\sqrt{\xi_i^2 + \xi_j^2}} \delta_{ij} + \frac{a \sin\varphi - x_{0j}}{\sqrt{\xi_i^2 + \xi_j^2}} \tag{72}$$

$\varphi$ is the angle between an observation point and the $x_i$ axis.

For $\boldsymbol{P} = P_z$,

$$\Phi_z = P_z \frac{\partial \phi}{\partial \eta}. \tag{73}$$

By substituting Equation (67) into Equations (71) and (73), we obtain

$$\Phi_j = P_j \Xi \left\{ A_m \frac{\partial J_m(\alpha\xi)}{\partial \xi} \cos(m\vartheta) e^{-ik_\eta \eta} - \frac{k_p^2}{\rho\omega^2} G_1 \frac{1}{1 - \left[\frac{r_{01}}{\alpha(a-r_0)}\right]^2} \frac{\partial J_m\left[\frac{r_{01}}{\alpha(a-r_0)}\xi\right]}{\partial \xi} \right\}, \tag{74}$$

$$\Phi_z = P_z \left\{ (-ik_\eta) A_m J_m(\alpha\xi) \cos(m\vartheta) e^{-ik_z \eta} - \frac{k_p^2}{\rho\omega^2} \frac{\partial G_1}{\partial \eta} \frac{1}{1 - \left[\frac{r_{01}}{\alpha(a-r_0)}\right]^2} J_m\left[\frac{r_{01}}{\alpha(a-r_0)}\xi\right] \right\}. \tag{75}$$

In Equation (75),

$$\frac{\partial G_1}{\partial \eta} = - \frac{J_0\left(\frac{r_{01}}{a-r_0}\xi_0\right)}{\pi(a-r_0)r_{01}\left[e^{-\frac{r_{01}z_0}{a-r_0}} + e^{-\frac{r_{01}(l-z_0)}{a-r_0}} - 2\right][J_1(r_{01})]^2}$$

$$\times \begin{cases} \left(\frac{r_{01}}{a-r_0}\right) e^{r_{01}\eta/(a-r_0)} & z_0 - l < \eta < 0 \\ \left(-\frac{r_{01}}{a-r_0}\right) e^{-r_{01}\eta/(a-r_0)} & 0 < \eta < l - z_0 \end{cases} . \tag{76}$$

From Equations (46) and (68), we obtain the SH potential:

$$X\hat{\boldsymbol{e}}_z = -\boldsymbol{\nabla} \times \boldsymbol{P}\chi$$

$$= -\left[ \left(\frac{\partial P_k\chi}{\partial \xi_j} - \frac{\partial P_j\chi}{\partial \eta}\right)\hat{\boldsymbol{i}} + \left(\frac{\partial P_i\chi}{\partial \eta} - \frac{\partial P_k\chi}{\partial \xi_i}\right)\hat{\boldsymbol{j}} + \left(\frac{\partial P_j\chi}{\partial \xi_i} - \frac{\partial P_i\chi}{\partial \xi_j}\right)\hat{\boldsymbol{e}}_z \right]$$

$$= -\left(\frac{\partial P_j\chi}{\partial \xi_i} - \frac{\partial P_i\chi}{\partial \xi_j}\right)\hat{\boldsymbol{e}}_z$$

For $\boldsymbol{P} = P_j$,

$$X_j = -P_j \left(\frac{\partial \chi}{\partial \xi_i} - \delta_{ji}\frac{\partial \chi}{\partial \xi_j}\right) = -P_j \left(\frac{\xi_i}{\xi} - \frac{\xi_j}{\xi}\delta_{ji}\right)\frac{\partial \chi}{\partial \xi} = -P_j \Sigma \frac{\partial \chi}{\partial \xi},$$

$$X_j = -P_j \Sigma \left\{ B_m \frac{\partial J_m(\beta\xi)}{\partial \xi} \sin(m\vartheta) e^{-ik_z \eta} - \frac{k_p^2}{\rho\omega^2} G_1 \frac{1}{1 - \left[\frac{r_{01}}{\beta(a-r_0)}\right]^2} \frac{\partial J_m\left[\frac{r_{01}}{\beta(a-r_0)}\xi\right]}{\partial \xi} \right\}. \tag{77}$$

In the above equation,

$$\Sigma = \frac{\xi_i}{\xi} - \frac{\xi_j}{\xi}\delta_{ij} = \frac{a \cos\varphi - x_{0i}}{\sqrt{\xi_i^2 + \xi_j^2}} - \frac{a \sin\varphi - x_{0j}}{\xi}\delta_{ij}. \tag{78}$$

For $\boldsymbol{P} = P_z$,

$$X_z = -P_z \left(\frac{\partial \chi}{\partial \xi_i}\delta_{iz} - \frac{\partial \chi}{\partial \xi_j}\delta_{jz}\right) = 0. \tag{79}$$

Similarly, for the SV potential, we find

$$\Psi_j = -P_j \Sigma C_m \frac{\partial J_m(\beta\xi)}{\partial\xi} \cos(m\vartheta)e^{-ik_z\eta}, \tag{80}$$

$$\Psi_z = 0. \tag{81}$$

All three potentials for the P, SH, and SV waves have now been completely determined, and we can thus derive the displacement components of $u_r, u_\theta$ and $u_z$ in terms of the three potentials. The relations between the displacements and potentials are given in Appendix A. First, let us derive the components generated by $P_j$. It can be easily shown that by substituting Equations (74), (77), and (80) into Equations (A5)–(A7), and taking the location of the point source as the origin ($\xi = 0$, $\vartheta = 0, \eta = 0$), the displacement $d$ component is reduced to

$$u_{dj} = P_j\left(A_{mj}F_{dj}^1 + B_{mj}F_{dj}^2 + C_{mj}F_{dj}^3 + F_{dj}^4\right)e^{-i\omega t}, \tag{82}$$

where subscript $d$ represents the radial ($r$), the tangential ($\theta$), or the axial ($z$) components.

For the radial component $u_{rj}$,

$$F_{rj}^1 = \left[\Xi\frac{\partial^2 J_m(\alpha\xi)}{\partial\xi^2}\right]\cos(m\vartheta)e^{-ik_z\eta}, \tag{83}$$

$$F_{rj}^2 = \frac{m\Sigma}{\xi}\frac{\partial J_m(\beta\xi)}{\partial\xi}\cos(m\vartheta)e^{-ik_z\eta}, \tag{84}$$

$$F_{rj}^3 = ik_z a\Sigma\left[\frac{\partial^2 J_m(\beta\xi)}{\partial\xi^2}\cos(m\vartheta)\right]e^{-ik_z\eta}, \tag{85}$$

$$F_{rj}^4 = -\Xi\left(\frac{k_p^2}{\rho\omega^2}\right)G_1\frac{1}{1-\left[\frac{r_{01}}{\alpha(a-r_0)}\right]^2}\left[\frac{\partial^2 J_m\left[\frac{r_{01}}{\alpha(a-r_0)}\xi\right]}{\partial\xi^2}\right]. \tag{86}$$

For the tangential component $u_{\theta j}$,

$$F_{\theta j}^1 = -\frac{m\Xi}{\xi}\frac{\partial J_m(\alpha\xi)}{\partial\xi}\sin(m\vartheta)e^{-ik_z\eta}, \tag{87}$$

$$F_{\theta j}^2 = -\left[\Sigma\frac{\partial^2 J_m(\beta\xi)}{\partial\xi^2}\right]\sin(m\vartheta)e^{-ik_z\eta}, \tag{88}$$

$$F_{\theta j}^3 = -\frac{ik_z am\Sigma}{\xi}\frac{\partial J_m(\beta\xi)}{\partial\xi}\sin(m\vartheta)e^{-ik_z\eta}, \tag{89}$$

$$F_{\theta j}^4 = -\Sigma\left(\frac{k_s^2}{\rho\omega^2}\right)\left\{G_1\frac{1}{1-\left[\frac{r_{01}}{\beta(a-r_0)}\right]^2}\left[\frac{\partial^2 J_m\left[\frac{r_{01}}{\beta(a-r_0)}\xi\right]}{\partial\xi^2}\right]\right\}. \tag{90}$$

For the axial component $u_{zj}$,

$$F_{zj}^1 = -ik_z\,\Xi\frac{\partial J_m(\alpha\xi)}{\partial\xi}\cos(m\vartheta)e^{-ik_z\eta}, \tag{91}$$

$$F_{zj}^2 = 0, \tag{92}$$

$$F_{zj}^3 = -a\Sigma\left\{\frac{\partial^3 J_m(\beta\xi)}{\partial\xi^3} + \frac{1}{\xi}\frac{\partial^2 J_m(\beta\xi)}{\partial\xi^2} - \frac{m^2}{\xi^2}\frac{\partial J_m(\beta\xi)}{\partial\xi}\right\}\cos(m\vartheta)e^{-ik_z\eta}, \tag{93}$$

$$F_{zj}^4 = -\Xi \left(\frac{k_p^2}{\rho\omega^2}\right) \left\{ \frac{\partial G_1}{\partial \eta} \frac{1}{1 - \left[\frac{r_{01}}{\alpha(a-r_0)}\right]^2} \left[ \frac{\partial J_m \left[\frac{r_{01}}{\alpha(a-r_0)}\xi\right]}{\partial \xi} \right] \right\}. \tag{94}$$

Following similar procedures, the three displacements generated by $P_z$ can be derived by substituting Equations (75), (79), and (81) into Equations (A6)–(A8):

$$u_{dz} = P_z \left( A_{mz} F_{dz}^1 + B_{mz} F_{dz}^2 + C_{mz} F_{dz}^3 + F_{dz}^4 \right) e^{-i\omega t} \tag{95}$$

For the radial component $u_{rz}$,

$$F_{rz}^1 = -ik_z \frac{\partial J_m(\alpha\xi)}{\partial \xi} \cos(m\theta) e^{-ik_z\eta}, \tag{96}$$

$$F_{rz}^2 = F_{rz}^3 = 0, \tag{97}$$

$$F_{rz}^4 = -\left(\frac{k_p^2}{\rho\omega^2}\right) \left\{ \frac{\partial G_1}{\partial \eta} \frac{1}{1 - \left[\frac{r_{01}}{\alpha(a-r_0)}\right]^2} \left[ \frac{\partial J_m \left[\frac{r_{01}}{\alpha(a-r_0)}\xi\right]}{\partial \xi} \right] \right\}. \tag{98}$$

For the tangential component $u_{\theta z}$,

$$F_{\theta z}^1 = \frac{ik_z m}{\xi} J_m(\alpha\xi) \sin(m\vartheta) e^{-ik_z\eta}, \tag{99}$$

$$F_{\theta z}^2 = F_{\theta z}^3 = F_{\theta z}^4 = 0. \tag{100}$$

For the axial component $u_{zz}$,

$$F_{zz}^1 = -k_z^2 J_m(\alpha\xi) \cos(m\vartheta) e^{-ik_z\eta}, \tag{101}$$

$$F_{zz}^2 = F_{zz}^3 = 0, \tag{102}$$

$$F_{zz}^4 = -\left(\frac{k_p^2}{\rho\omega^2}\right) \left\{ \frac{\partial^2 G_1}{\partial \eta^2} \frac{1}{1 - \left[\frac{r_{01}}{\alpha(a-r_0)}\right]^2} \left[ \frac{\partial J_m \left[\frac{r_{01}}{\alpha(a-r_0)}\xi\right]}{\partial \xi} \right] \right\}. \tag{103}$$

The only remaining task to complete the displacement fields is to determine the coupling constants $A_m$, $B_m$, and $C_m$. These constants can be determined directly by applying a fundamental set of linear elastic boundary problems. The outer surface of the cylinder studied in the present paper is stress-free. Thus, the following stress components are zero under these circumstances, i.e., $\xi = a - r_0$:

$$\sigma_{rr} = \sigma_{r\theta} = \sigma_{rz} = 0. \tag{104}$$

Substituting Equations (A17)–(A21) into Equation (104) yields the following algebraic equations

$$\begin{bmatrix} a_{11f} & a_{12f} & a_{13f} \\ a_{21f} & a_{22f} & a_{23f} \\ a_{31f} & a_{32f} & a_{33f} \end{bmatrix} \begin{bmatrix} A_{mf} \\ B_{mf} \\ C_{mf} \end{bmatrix} = \begin{bmatrix} b_{1f} \\ b_{2f} \\ b_{3f} \end{bmatrix}, \tag{105}$$

where $f = j$ for $P_j$ and $f = z$ for $P_z$. The elements in Equation (105) are given in Appendix A. For $P_j$, the elements of $b_{1j}$, $b_{2j}$ and $b_{3j}$ are non-zero; therefore, the coupling constants can be determined by solving Equation (105).

The CF in Equations (1) and (43) is the force impulse defined as

$$\boldsymbol{P}(t) = P_0 t \exp(-bt), \tag{106}$$

where $P_0$ and $b$ are parameters that determine the amplitude and the duration of the wave, respectively [29]. The arrival time $\tau$ of the impulse at the position $(\xi_i, \xi_j, \eta)$ must be considered, since the force impulse is acting in the $f$ direction at position $\xi_i = \xi_j = 0$ and $\eta = 0$ at $t = 0$. Replacing $t$ with $(t - \tau)$ yields

$$P_f(t - \tau) = P_{0,f}(t - \tau)e^{-b(t-\tau)}. \tag{107}$$

The displacement components generated by the $\Phi_f$ potential, and the $X_f$ and $\Psi_f$ potentials, correspond to the compressional (P), and the shear (SH and SV) waves, respectively; therefore, the arrival times of the two waves are given as

$$\tau_P = \frac{\sqrt{\xi_i^2 + \xi_j^2 + \eta^2}}{c_P}, \quad \tau_S = \frac{\sqrt{\xi_i^2 + \xi_j^2 + \eta^2}}{c_S}, \tag{108}$$

where $c_P$ and $c_S$ are the velocities of the P and S waves, respectively. The displacements generated by $P_f$ can be rewritten as

$$u_{rf} = \left[ P_f(t - \tau_P) \left( A_{mf} F_{rf}^1 + F_{rf}^4 \right) + P_f(t - \tau_s) \left( B_{mf} F_{rf}^2 + C_{mf} F_{rf}^3 \right) \right] e^{-i\omega t}, \tag{109}$$

$$u_{\theta f} = \left[ P_f(t - \tau_P) \left( A_{mf} F_{\theta j}^1 \right) + P_f(t - \tau_s) \left( B_{mf} F_{\theta f}^2 + C_{mf} F_{\theta f}^3 + F_{\theta f}^4 \right) \right] e^{-i\omega t}, \tag{110}$$

$$u_{zf} = \left[ P_f(t - \tau_P) \left( A_{mf} F_{zf}^1 + F_{zf}^4 \right) + P_f(t - \tau_s) \left( B_{mf} F_{zf}^2 \right) \right] e^{-i\omega t}. \tag{111}$$

Furthermore, the P, SH, and SV waves can be obtained from Equations (109)–(111) as

$$u_f^P = P_f(t - \tau_P) \left( A_{mf} F_{df}^1 + F_{rf}^4 \right) e^{-i\omega t} \quad (d = r, \ z \text{ and } \theta), \tag{112}$$

$$u_f^{SV} = P_f(t - \tau_S) \left( C_{mf} F_{df}^3 \right) e^{-i\omega t} \quad (d = r, \ z \text{ and } \theta), \tag{113}$$

$$u_f^{SH} = P_f(t - \tau_S) \left( B_{mf} F_{df}^2 + F_{\theta f}^4 \right) e^{-i\omega t} \quad (d = r, \ z \text{ and } \theta). \tag{114}$$

## 4. Simulation

In this study, we consider only the azimuthal dependence of the wave propagation arising entirely from the direction of the force vector, $m = 0$. A stainless-steel (SS) cylinder ($a = 0.50$ m, $l = 2.0$ m, $\rho = 7.80 \times 10^3$ kg/m$^3$, $c_P = 5.98$ km/s, and $c_S = 3.30$ km/s) was used as the test specimen. Stiffness parameters ($c_{11} = c_{22} = c_{33} = 2.086 \times 10^{11}$, $c_{12} = c_{13} = c_{23} = 1.465 \times 10^{11}$, $c_{44} = c_{55} = c_{66} = 1.269 \times 10^{11}$ N m$^{-2}$) can be found for austenitic stainless steel [30]. Previously, the natural frequencies of SS active on AE in the range of 0–300 kHz were determined experimentally by a tensile test (Appendix B). The frequency of $\nu = 155.4$ kHz (the angular frequency $\omega = 2\pi\nu$) was found to be predominant.

First, we determined $k_z$ by applying the boundary conditions to Equations (A20) and (A22) which were associated with the displacements due to $P_z$. From Equation (105), two solutions of $A_0$ were obtained, as follows:

$$A_{0z}^1 = \frac{b_{1z}}{a_{11z}} \quad \text{or} \quad A_{0z}^2 = \frac{b_{3z}}{a_{31z}} \tag{115}$$

since $a_{13z} = a_{33z} = 0$.

When $m = 0$,

$$\begin{aligned} a_{11z} &= ik_z \left[ c_{13}k_z^2 J_0(\alpha\xi) - \frac{c_{12}}{\xi} \frac{\partial J_0(\alpha\xi)}{\partial \xi} - c_{11} \frac{\partial^2 J_0(\alpha\xi)}{\partial \xi^2} \right] e^{-ik_z\eta} \\ &= ik_z \left\{ c_{13}k_z^2 J_0(\alpha\xi) + \frac{c_{12}}{\xi} \alpha J_1(\alpha\xi) - c_{11} 2 \left( \frac{\alpha}{2} \right)^2 [-J_0(\alpha\xi) + J_2(\alpha\xi)] \right\} e^{-ik_z\eta}, \end{aligned} \tag{116}$$

$$b_{1z} = -\left(\frac{k_p^2}{\rho\omega^2}\right)\left(\frac{1}{1-\left[\frac{r_{01}}{\alpha(a-r_0)}\right]^2}\right)\left\{c_{11}\frac{\partial G_1}{\partial\eta}\frac{\partial^2 J_0\left[\frac{r_{01}}{\alpha(a-r_0)}\xi\right]}{\partial\xi^2} + c_{12}\frac{1}{\xi}\frac{\partial G_1}{\partial\eta}\frac{\partial J_0\left[\frac{r_{01}}{\alpha(a-r_0)}\xi\right]}{\partial r}\right.$$
$$\left. + c_{13}\frac{\partial^3 G_1}{\partial\eta^3}J_0\left[\frac{r_{01}}{\alpha(a-r_0)}\xi\right]\right\} \tag{117}$$

$$= -\left(\frac{k_p^2}{\rho\omega^2}\right)\left(\frac{1}{1-\left[\frac{r_{01}}{\alpha(a-r_0)}\right]^2}\right)\left\langle c_{11}\frac{\partial G_1}{\partial\eta}2\left[\frac{r_{01}}{2\alpha(a-r_0)}\right]^2\left\{-J_0\left[\frac{r_{01}}{\alpha(a-r_0)}\xi\right] + J_2\left[\frac{r_{01}}{\alpha(a-r_0)}\xi\right]\right\}\right.$$
$$\left. -c_{12}\frac{1}{\xi}\frac{\partial G_1}{\partial\eta}\frac{r_{01}}{\alpha(a-r_0)}J_1\left[\frac{r_{01}}{\alpha(a-r_0)}\xi\right] + c_{13}\frac{\partial^3 G_1}{\partial\eta^3}J_0\left[\frac{r_{01}}{\alpha(a-r_0)}\xi\right]\right\rangle,$$

$$a_{31z} = c_{44}k_z^2\alpha J_1(\alpha\xi)e^{-ik_z\eta}, \tag{118}$$

$$b_{3z} = -c_{44}\left(\frac{k_p^2}{\rho\omega^2}\right)\left\{\frac{\partial^2 G_1}{\partial\eta^2}\frac{1}{1-\left[\frac{r_{01}}{\alpha(a-r_0)}\right]^2}2\frac{\partial J_0\left[\frac{r_{01}}{\alpha(a-r_0)}\xi\right]}{\partial\xi}\right\} \tag{119}$$
$$= 2c_{44}\left(\frac{k_p^2}{\rho\omega^2}\right)\frac{\partial^2 G_1}{\partial\eta^2}\frac{1}{1-\left[\frac{r_{01}}{\alpha(a-r_0)}\right]^2}\frac{r_{01}}{\alpha(a-r_0)}J_1\left[\frac{r_{01}}{\alpha(a-r_0)}\xi\right].$$

In Equations (117) and (119), the partial derivatives of $G_1$ with respect to $\eta$ can easily be obtained from Equation (42). On the circumference, $\xi = a - r_0$ and $\eta = l - z_0$, we solved the roots of the function

$$f(k_z) = \frac{b_{1z}}{a_{11z}} - \frac{b_{3z}}{a_{31z}} = 0, \tag{120}$$

as a function of the shortest distance from the point source to the end plate of the cylinder ($\eta$) at a given $\xi$. It should be noted that $f(k_z)$ is independent of $\Xi$ and $\Sigma$. Figure 2 shows the $\eta$ dependence of three roots (n = 1–3) of Equation (120) for $r_0 = 0$ $(x_{0i} = x_{0j} = 0 \text{ m})$ and 0.25 $(x_{0i} = 0, x_{0j} = 0.25 \text{ m})$. In the simulation, we selected the first root (n = 1) at the given $r_0$ and $z_0$ values.

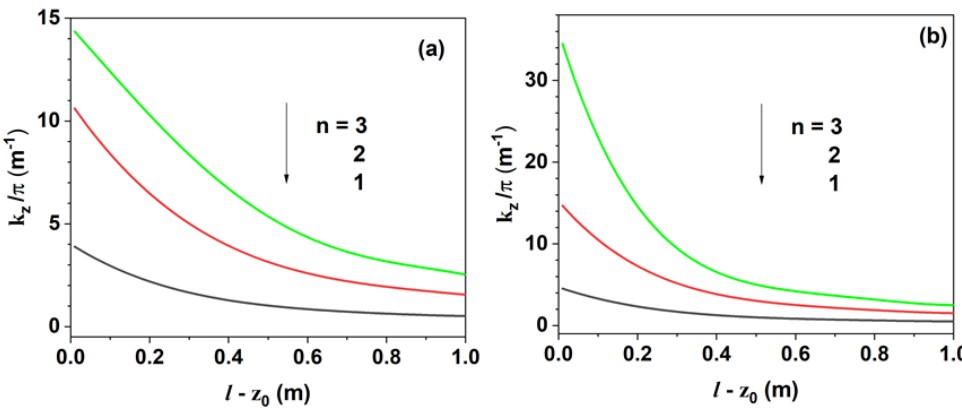

**Figure 2.** Spectra of three sets of $k_z$ values calculated as a function of $\eta(l - z_0)$: (**a**) $r_0 = 0$ $\left(x_{0i} = x_{0j} = 0, z_0 = 1 \text{ m}\right)$ and (**b**) $r_0 = 0.25 \text{ m}$ $\left(x_{0i} = 0, x_{0j} = 0.25, z_0 = 1 \text{ m}\right)$.

Next, we determined $P_0$ and $b$ in Equation (107). The envelope of a given wave was evaluated by fitting its normalized form to determine $b$. As shown in Figure 3, when $b = 6.0 \times 10^4 \text{ s}^{-1}$, the duration ($\Delta\tau$) of the wave was approximately 1 ms. By increasing the $b$ value, the duration of the envelope decreased exponentially at $b = 4.0 \times 10^5 \text{ s}^{-1}$ with $\Delta\tau \cong 25 \text{ μs}$. In the simulation, the force exerted by the point source was fixed to 1 N with $P_0 = 1.0 \times 10^{10} \text{ N s}^{-1}$ and $b = 1.0 \times 10^5 \text{ s}^{-1}$.

In the simulation two positions of the point source of 1 N were considered, with coordinates of (0 m, 0 m, 1 m) and (0 m, 0.25 m, 1 m). Figure 4a shows the displacements, and their wave properties, at the (0.5 m, 45°, 1 m) position, generated by the $P_f$ point source located at the center of the cylinder (0 m, 0 m, 1 m). The $P_j$ excitation produces an

axial displacement that is stronger than the radial and tangential displacements. These displacements result in the P wave being the main wave, with a minor SH wave and very weak SV wave (Figure 4b). As shown in Figure 4c, the displacement features generated by the $P_z$ excitation differ significantly from those generated by the $P_j$ excitation. For the $P_j$ excitation, the maximum values of $u_{rj}$ and $u_{zj}$ at the (0.5 m, 45°, 1 m) position were 0.29 and 0.71 nm, respectively, whereas for the $P_z$ excitation, they were 28.1 and 5.0 nm, respectively. The amplitudes of the displacements due to the $P_z$ excitation were much stronger than those due to the $P_j$ excitation. It should be noted that the $P_z$ excitation produces only the P wave (Figure 4d). The angular dependence of the displacement was also calculated (Figure 5). When the point source is located at the center of the circular plane, the angular dependences of $u_{rj}$, $u_{zj}$ and $u_{tj}$ arise from only $\Xi$ and $\Sigma$ as defined in Equations (72) and (78), respectively. For $P_z$ excitation, the displacements of $u_{rz}$ and $u_{zz}$ are free from these factors. When the distances from the point source to the circumference are not equivalent, the angular dependences of the radial and axial displacements are highly significant. These effects are due not only to $\Xi$ and $\Sigma$ but also the superposition of the Bessel functions involved in the displacement equations. At a certain angle, some $a_{ij}$ values in Equation (105) become too small to cause the sudden increase in displacement.

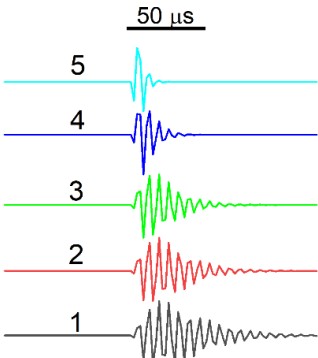

**Figure 3.** Effect of the $b$ parameter of the point source on the envelopes of the simulated $u_{rj}$ waveforms: (1) $6 \times 10^4$, (2) $8 \times 10^4$, (3) $1 \times 10^5$, (4) $2 \times 10^5$, and (5) $4 \times 10^5$.

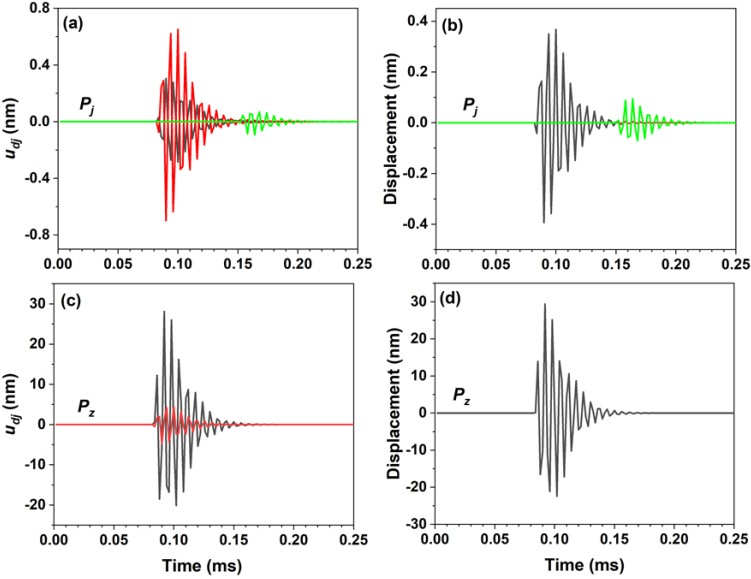

**Figure 4.** Displacements at the (0.5 m, 45°, 1 m) position generated by the point source of 1.0 N located at the center of the cylinder: (**a**,**c**) radial (black line), axial (red line), and tangential (green line) components; and (**b**,**d**) compression P (black line) wave and vertically polarized (SV, red line) and horizontally polarized (SH, green line) shear waves.

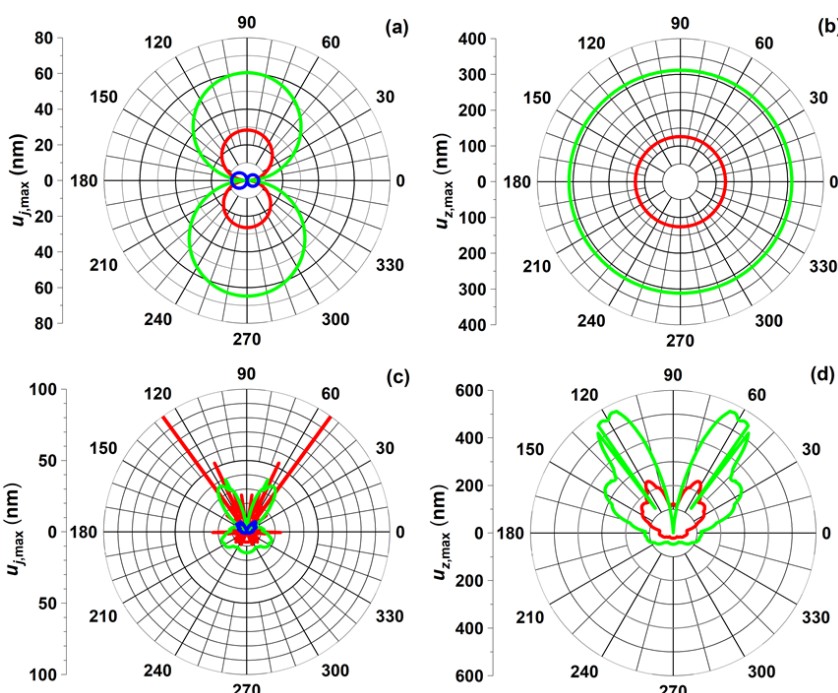

**Figure 5.** Angular dependences of the maximum displacements at the (0.5 m, 45°, 1 m) position: radial (red line), axial (green line), and tangential (blue line) displacements generated by (**a,c**) $P_j$ and (**b,d**) $P_z$ of 1.0 N, located at (**a,b**) (0 m, 0 m, 1 m) and (**c,d**) (0 m, 0.25 m, 1 m) positions.

## 5. Conclusions

In this paper, we present a mathematical model for AE generated by a point source in a TIC. The point source as the CF vector is a model for an initiation state of crack formation in a solid medium. In the cylindrical system, the displacement field vector is very complex: compressional (P), and vertical and horizontal shear (SV and SH) wave potentials are coupled. Introducing CFIPs into the NL equation leads to the solutions of the radial, tangential, and axial displacements in the cylindrical system. The solutions obtained in this study can be used for analyzing experimental data for the NDT of cylinders. In the near future, similar mathematical models will be developed for shells and multilayered cylinders.

**Author Contributions:** Conceptualization, methodology and investigation, K.B.K. and J.-G.K.; software, validation, and formal analysis, K.B.K.; resources and data curation, S.G.L.; writing—original draft preparation, K.B.K. and J.-G.K.; writing—review and editing, visualization, and supervision, J.-G.K.; project administration and funding acquisition, B.K.K. All authors have read and agreed to the published version of the manuscript.

**Funding:** This research was funded by the Korea Institute of Energy Technology Evaluation and Planning (KETEP) and the Ministry of Trade, Industry & Energy (MOTIE) of the Republic of Korea (NO. 20202910100070).

**Institutional Review Board Statement:** Not applicable.

**Informed Consent Statement:** Not applicable.

**Data Availability Statement:** Not applicable.

**Conflicts of Interest:** The authors declare no conflict of interest.

## Appendix A

Some gradient, divergence and curl operators on the potentials are as follows:

$$\boldsymbol{\nabla}\Phi = \frac{\partial\Phi}{\partial r}\hat{\boldsymbol{r}} + \frac{1}{r}\frac{\partial\Phi}{\partial\theta}\hat{\boldsymbol{\theta}} + \frac{\partial\Phi}{\partial z}\hat{\boldsymbol{z}}, \tag{A1}$$

$$\boldsymbol{\nabla} \times (X\hat{e}_z) = \left[\frac{1}{r}\frac{\partial(X\hat{e}_z)_z}{\partial\theta} - \frac{\partial(X\hat{e}_z)_\theta}{\partial z}\right]\hat{\boldsymbol{r}} + \left[\frac{\partial(X\hat{e}_z)_r}{\partial z} - \frac{\partial(X\hat{e}_z)_z}{\partial r}\right]\hat{\boldsymbol{\theta}}$$
$$+ \frac{1}{r}\left\{\frac{\partial[r(X\hat{e}_z)_\theta]}{\partial r} - \frac{\partial(X\hat{e}_z)_r}{\partial\theta}\right\}\hat{\boldsymbol{z}} = \frac{1}{r}\frac{\partial X}{\partial\theta}\hat{\boldsymbol{r}} - \frac{\partial X}{\partial r}\hat{\boldsymbol{\theta}},$$
(A2)

$$\boldsymbol{\nabla} \times \boldsymbol{\nabla} \times (\Psi\hat{e}_z) = \boldsymbol{\nabla} \times \left(\frac{1}{r}\frac{\partial\Psi}{\partial\theta}\hat{\boldsymbol{r}} - \frac{\partial\Psi}{\partial r}\hat{\boldsymbol{\theta}}\right)$$
$$= \left[-\frac{\partial\left(-\frac{\partial\Psi}{\partial r}\right)}{\partial z}\right]\hat{\boldsymbol{r}} + \left[\frac{\partial\left(\frac{1}{r}\frac{\partial\Psi}{\partial\theta}\right)}{\partial z}\right]\hat{\boldsymbol{\theta}} + \frac{1}{r}\left[\frac{\partial\left[r\left(-\frac{\partial\Psi}{\partial r}\right)\right]}{\partial r} - \frac{\partial\left(\frac{1}{r}\frac{\partial\Psi}{\partial\theta}\right)}{\partial\theta}\right]\hat{\boldsymbol{z}}$$
$$= \left(\frac{\partial^2\Psi}{\partial r\partial z}\right)\hat{\boldsymbol{r}} + \left(\frac{1}{r}\frac{\partial^2\Psi}{\partial\theta\partial z}\right)\hat{\boldsymbol{\theta}} - \frac{1}{r}\left(\frac{\partial\Psi}{\partial r} + r\frac{\partial^2\Psi}{\partial r^2} + \frac{1}{r}\frac{\partial^2\Psi}{\partial\theta^2}\right)\hat{\boldsymbol{z}}$$
(A3)

Substituting the above equations into Equation (44) gives

$$\boldsymbol{u} = \left(\frac{\partial\Phi}{\partial r} + \frac{1}{r}\frac{\partial X}{\partial\theta} + a\frac{\partial^2\Psi}{\partial r\partial z}\right)\hat{\boldsymbol{r}} + \left(\frac{1}{r}\frac{\partial\Phi}{\partial\theta} + \frac{\partial X}{\partial r} + a\frac{1}{r}\frac{\partial^2\Psi}{\partial\theta\partial z}\right)\hat{\boldsymbol{\theta}}$$
$$+ \left[\frac{\partial\Phi}{\partial z} - a\left(\frac{\partial^2\Psi}{\partial r^2} + \frac{1}{r}\frac{\partial\Psi}{\partial r} + \frac{1}{r^2}\frac{\partial^2\Psi}{\partial\theta^2}\right)\right]\hat{\boldsymbol{z}}.$$
(A4)

From Equation (A5), we obtain the displacement components as

$$u_r = \frac{\partial\Phi}{\partial r} + \frac{1}{r}\frac{\partial X}{\partial\theta} + a\frac{\partial^2\Psi}{\partial r\partial z},$$
(A5)

$$u_\theta = \frac{1}{r}\frac{\partial\Phi}{\partial\theta} + \frac{\partial X}{\partial r} + a\frac{1}{r}\frac{\partial^2\Psi}{\partial\theta\partial z},$$
(A6)

$$u_z = \frac{\partial\Phi}{\partial z} - a\left(\frac{\partial^2\Psi}{\partial r^2} + \frac{1}{r}\frac{\partial\Psi}{\partial r} + \frac{1}{r^2}\frac{\partial^2\Psi}{\partial\theta^2}\right),$$
(A7)

in which $u_r$, $u_\theta$ and $u_z$ denote the radial, tangential, and axial displacements, respectively. The radial component can be obtained by substituting Equations (71), (77), and (79) into Equation (A6) in the $(\xi, \vartheta, \eta)$ coordinates.

$$u_{rj} = \frac{\partial\Phi_j}{\partial\xi} + \frac{1}{\xi}\frac{\partial X_j}{\partial\vartheta} + a\frac{\partial^2\Psi_j}{\partial\xi\partial\eta}$$
$$= P_j\left\langle \Xi\left\{A_m\cos(m\vartheta)e^{-ik_z\eta}\frac{\partial^2 J_m(\alpha\xi)}{\partial\xi^2} - \left(\frac{k_p^2}{\rho\omega^2}\right)G_1\frac{1}{1 - \left[\frac{r_{01}}{\alpha(a-r_0)}\right]^2}\frac{\partial^2 J_m\left[\frac{r_{01}}{\alpha(a-r_0)}\xi\right]}{\partial\xi^2}\right\}\right.$$
$$\left. + \frac{m\Sigma}{\xi}\left[B_m\frac{\partial J_m(\beta\xi)}{\partial\xi}\cos(m\vartheta)e^{-ik_z\eta}\right] + a(ik_z)\Sigma C_m\left[\frac{\partial^2 J_m(\beta\xi)}{\partial\xi^2}\cos(m\vartheta)\right]\right\rangle e^{-ik_z\eta}e^{-i\omega t},$$
(A8)

where $\frac{\partial\Xi}{\partial\xi} \cong 0$ and $\frac{\partial\Sigma}{\partial\xi} \cong 0$. By rearranging the above equation in terms of the $A_m$, $B_m$ and $C_m$ coupling constants, we obtain

$$u_{rj} = P_j\left[\sum_{m=0}^{\infty}\left(A_m F_{rj}^1 + B_m F_{rj}^2 + C_m F_{rj}^3\right) + F_{rj}^4\right]e^{-i\omega t},$$
(A9)

where

$$F_{rj}^1 = \left[\Xi\frac{\partial^2 J_m(\alpha\xi)}{\partial\xi^2}\right]\cos(m\vartheta)e^{-ik_z\eta},$$
(A10)

$$F_{rj}^2 = \frac{m}{\xi}\frac{\partial J_m(\beta\xi)}{\partial\xi}\Sigma\cos(m\vartheta)e^{-ik_z\eta},$$
(A11)

$$F_{rj}^3 = ik_z a\Sigma\left[\frac{\partial^2 J_m(\beta\xi)}{\partial\xi^2}\cos(m\vartheta)\right]e^{-ik_z\eta},$$
(A12)

$$F_{rj}^4 = -\left(\frac{k_p^2}{\rho\omega^2}\right)\Xi\left[G_1\frac{1}{1 - \left[\frac{r_{01}}{\alpha(a-r_0)}\right]^2}\frac{\partial^2 J_m\left[\frac{r_{01}}{\alpha(a-r_0)}\xi\right]}{\partial r^2}\right].$$
(A13)

For TIC, the stress-strain [18] and the strain-displacement [31] relations are given as

$$
\begin{bmatrix} \sigma_{rr} \\ \sigma_{\theta\theta} \\ \sigma_{zz} \\ \sigma_{\theta z} \\ \sigma_{rz} \\ \sigma_{r\theta} \end{bmatrix} =
\begin{bmatrix}
c_{11} & c_{12} & c_{13} & 0 & 0 & 0 \\
c_{12} & c_{11} & c_{13} & 0 & 0 & 0 \\
c_{13} & c_{13} & c_{33} & 0 & 0 & 0 \\
0 & 0 & 0 & c_{44} & 0 & 0 \\
0 & 0 & 0 & 0 & c_{44} & 0 \\
0 & 0 & 0 & 0 & 0 & (c_{11}-c_{12})/2
\end{bmatrix}
\begin{bmatrix} \epsilon_{rr} \\ \epsilon_{\theta\theta} \\ \epsilon_{zz} \\ 2\epsilon_{\theta z} \\ 2\epsilon_{rz} \\ 2\epsilon_{r\theta} \end{bmatrix},
$$

and

$$
\epsilon_{rr} = \frac{\partial u_r}{\partial r}, \; \epsilon_{\theta\theta} = \frac{1}{r}\frac{\partial u_\theta}{\partial \theta} + \frac{u_r}{r}, \; \epsilon_{\theta z} = \frac{1}{2}\left(\frac{\partial u_\theta}{\partial z} + \frac{1}{r}\frac{\partial u_z}{\partial \theta}\right)
$$
$$
\epsilon_{zz} = \frac{\partial u_z}{\partial z}, \; \epsilon_{r\theta} = \frac{1}{2}\left(\frac{1}{r}\frac{\partial u_r}{\partial \theta} + \frac{\partial u_\theta}{\partial r} - \frac{u_\theta}{r}\right), \; \epsilon_{rz} = \frac{1}{2}\left(\frac{\partial u_z}{\partial r} + \frac{\partial u_r}{\partial z}\right),
$$

respectively. From these relations, one can obtain three stresses of $\sigma_{rr}$, $\sigma_{r\phi}$ and $\sigma_{rz}$ as [18]:

$$
\sigma_{rr} = c_{11}\left(\frac{\partial u_r}{\partial r}\right) + c_{12}\left(\frac{u_r}{r} + \frac{1}{r}\frac{\partial u_\theta}{\partial \theta}\right) + c_{13}\left(\frac{\partial u_z}{\partial z}\right), \tag{A14}
$$

$$
\sigma_{r\theta} = \frac{(c_{11}-c_{12})}{2}\left(\frac{\partial u_\theta}{\partial r} - \frac{u_\theta}{r} + \frac{1}{r}\frac{\partial u_r}{\partial \theta}\right), \tag{A15}
$$

$$
\sigma_{rz} = c_{44}\left(\frac{\partial u_z}{\partial r} + \frac{\partial u_r}{\partial z}\right) \tag{A16}
$$

From $u_{rj}$, $u_{\theta j}$ and $u_{zj}$

$$
\sigma_{rrj} = P_j\left(a_{11j}A_{mj} + a_{12j}B_{mj} + a_{12j}C_{mj} + b_{1j}\right)e^{-i\omega t}, \tag{A17}
$$

where

$$
a_{11j} = c_{11}\frac{\partial F^1_{rj}}{\partial \xi} + c_{12}\frac{F^1_{rj}}{\xi} + c_{12}\frac{1}{\xi}\frac{\partial F^1_{\theta j}}{\partial \vartheta} + c_{13}\left(\frac{\partial F^1_{zj}}{\partial \eta}\right)
$$
$$
= \Xi\left\{-\left(c_{12}\frac{m^2}{\xi^2} + c_{13}k_z^2\right)\frac{\partial J_m(\alpha\xi)}{\partial \xi} + c_{12}\frac{1}{\xi}\frac{\partial^2 J_m(\alpha\xi)}{\partial \xi^2} + c_{11}\frac{\partial^3 J_m(\alpha\xi)}{\partial \xi^3}\right\}\cos(m\vartheta)e^{-ik_z\eta},
$$

$$
a_{12j} = c_{11}\frac{\partial F^2_{rj}}{\partial \xi} + c_{12}\frac{F^2_{rj}}{\xi} + c_{12}\frac{1}{\xi}\frac{\partial F^2_{\theta j}}{\partial \vartheta}
$$
$$
= (-c_{11} + c_{12})\frac{m\Sigma}{\xi}\left[\frac{1}{\xi}\frac{\partial J_m(\beta\xi)}{\partial \xi} - \frac{\partial^2 J_m(\beta\xi)}{\partial \xi^2}\right]\cos(m\vartheta)e^{-ik_z\eta},
$$

$$
a_{13j} = c_{11}\frac{\partial F^3_{rj}}{\partial r} + c_{12}\frac{F^3_{rj}}{r} + c_{12}\frac{1}{r}\frac{\partial F^3_{\theta j}}{\partial \vartheta} + c_{13}\frac{\partial F^3_{zj}}{\partial z}
$$
$$
= ik_z a\Sigma\left\{(c_{12} + c_{13})\frac{1}{\xi}\left[-\frac{m^2}{\xi}\frac{\partial J_m(\beta\xi)}{\partial \xi} + \frac{\partial^2 J_m(\beta\xi)}{\partial \xi^2}\right] + (c_{11} + c_{13})\frac{\partial^3 J_m(\beta\xi)}{\partial \xi^3}\right\}e^{-ik_z\eta},
$$

$$
b_{1j} = c_{11}\frac{\partial F^4_{rj}}{\partial r} + c_{12}\frac{F^4_{rj}}{r} + c_{13}\frac{\partial F^4_{zj}}{\partial \eta}
$$
$$
= -\left(\frac{k_p^2}{\rho\omega^2}\right)\Xi\frac{1}{1-\left[\frac{r_{01}}{\alpha(a-r_0)}\right]^2}\left\{c_{11}G_1\frac{\partial^3 J_0\left[\frac{r_{01}}{\alpha(a-r_0)}\xi\right]}{\partial \xi^3} + c_{12}\frac{1}{\xi}G_1\frac{\partial^2 J_0\left[\frac{r_{01}}{\alpha(a-r_0)}\xi\right]}{\partial \xi^2} + c_{13}\frac{\partial^2 G_n}{\partial \eta^2}\frac{\partial J_0\left[\frac{r_{01}}{\alpha(a-r_0)}\xi\right]}{\partial \xi}\right\},
$$

$$
\sigma_{r\theta j} = P_j\left(a_{21j}A_{mj} + a_{22j}B_{mj} + a_{23j}C_{mj} + b_{2j}\right)e^{-i\omega t}, \tag{A18}
$$

where

$$
a_{21j} = \frac{(c_{11}-c_{12})}{2}\left(\frac{\partial F^1_{\theta j}}{\partial \xi} - \frac{F^1_{\theta j}}{\xi} + \frac{1}{\xi}\frac{\partial F^1_{rj}}{\partial \vartheta}\right)
$$
$$
= \frac{(c_{11}-c_{12})}{2}(2m\Xi)\left[\frac{1}{\xi^2}\frac{\partial J_m(\alpha\xi)}{\partial \xi} - \frac{1}{\xi}\frac{\partial^2 J_m(\alpha\xi)}{\partial \xi^2}\right]\sin(m\vartheta)e^{-ik_z\eta},
$$

$$a_{22j} = \frac{(c_{11}-c_{12})}{2}\left(\frac{\partial F_{\theta j}^2}{\partial r} - \frac{F_{\theta j}^2}{r} + \frac{1}{r}\frac{\partial F_{rj}^2}{\partial \theta}\right)$$

$$= -\frac{(c_{11}-c_{12})}{2}\Sigma\left[\frac{m^2}{\xi}\frac{\partial J_m(\beta\xi)}{\partial\xi} + \left(2-\frac{1}{\xi}\right)\frac{\partial^2 J_m(\beta\xi)}{\partial\xi^2} + \frac{\partial^3 J_m(\beta\xi)}{\partial\xi^3}\right]\sin(m\vartheta)e^{-ik_z\eta},$$

$$a_{23j} = \frac{(c_{11}-c_{12})}{2}\left(\frac{\partial F_{\theta j}^3}{\partial r} - \frac{F_{\theta j}^3}{r} + \frac{1}{r}\frac{\partial F_{rj}^3}{\partial\theta}\right)$$

$$= \frac{(c_{11}-c_{12})}{2}\left(\frac{ik_z am\Sigma}{\xi}\right)\left[\frac{2}{\xi}\frac{\partial^2 J_m(\beta\xi)}{\partial\xi^2} - \frac{\partial^2 J_m(\beta\xi)}{\partial\xi^2}\right]\sin(m\vartheta)e^{-ik_z\eta},$$

$$b_{2j} = \frac{(c_{11}-c_{12})}{2}\left(\frac{\partial F_{\theta j}^4}{\partial\xi} - \frac{F_{\theta j}^4}{\xi} + \frac{1}{\xi}\frac{\partial F_{rj}^4}{\partial\vartheta}\right)$$

$$= \frac{(c_{11}-c_{12})}{2}\left(-\frac{k_s^2}{\rho\omega^2}\right)\Sigma\frac{G_1}{1-\left[\frac{r_{01}}{\beta(a-r_0)}\right]^2}\left\{-\frac{1}{\xi}\frac{\partial^2 J_m\left[\frac{r_{01}}{\beta(a-r_0)}\xi\right]}{\partial\xi^2} + \frac{\partial^3 J_m\left[\frac{r_{01}}{\beta(a-r_0)}\xi\right]}{\partial\xi^3}\right\},$$

and

$$\sigma_{rzj} = (a_{31j}A_{mj} + a_{32j}B_{mj} + a_{33j}C_{mj} + b_{3j})e^{-i\omega t}, \tag{A19}$$

where

$$a_{31j} = c_{44}\left(\frac{\partial F_{zj}^1}{\partial\xi} + \frac{\partial F_{rj}^1}{\partial\eta}\right)$$

$$= -2c_{44}(ik_z\Xi)\frac{\partial^2 J_m(\alpha\xi)}{\partial\xi^2}\cos(m\vartheta)e^{-ik_z z},$$

$$a_{32j} = c_{44}\left(\frac{\partial F_{rj}^2}{\partial\eta}\right)$$

$$= -c_{44}\left(\frac{ik_z m\Sigma}{\xi}\right)\frac{\partial J_m(\beta\xi)}{\partial\xi}\cos(m\vartheta)e^{-ik_z\eta},$$

$$a_{33j} = c_{44}\left(\frac{\partial F_{zj}^3}{\partial\xi} + \frac{\partial F_{rj}^3}{\partial\eta}\right)$$

$$= c_{44}a\Sigma\left[\left(\frac{m^2}{\xi^2}\right)\left(1 - \frac{2}{\xi} - ik_z\right)\frac{\partial J_m(\beta\xi)}{\partial\xi} + \frac{1}{\xi}\left(\frac{1+m^2}{\xi} + ik_z\right)\frac{\partial^2 J_m(\beta\xi)}{\partial\xi^2}\right.$$
$$\left. - \left(\frac{1}{\xi} - ik_z\right)\frac{\partial^3 J_m(\beta\xi)}{\partial\xi^3} - \frac{\partial^4 J_m(\beta\xi)}{\partial\xi^4}\right]\cos(m\vartheta)e^{-ik_z\eta},$$

$$b_{3j} = c_{44}\left(\frac{\partial F_{zj}^4}{\partial\xi} + \frac{\partial F_{rj}^4}{\partial\eta}\right)$$

$$= -c_{44}\left(\frac{k_p^2}{\rho\omega^2}\right)\Xi\left\{\frac{1}{1-\left(\frac{r_{0n}}{\alpha a}\right)^2}\left[\frac{\partial G_1}{\partial\eta}\frac{\partial^2 J_m\left[\frac{r_{01}}{\alpha(a-r_0)}\xi\right]}{\partial\xi^2} + \frac{\partial^2 G_n}{\partial\eta^2}\frac{\partial J_m\left[\frac{r_{01}}{\alpha(a-r_0)}\xi\right]}{\partial\xi}\right]\right\}.$$

From $u_{rz}$, $u_{\theta z}$ and $u_{zz}$

$$\sigma_{rrz} = P_z(a_{11z}A_{mz} + a_{12z}B_{mz} + a_{12j}C_{mz} + b_{1z})e^{-i\omega t}, \tag{A20}$$

where

$$a_{11z} = ik_z\left[\left(c_{12}\frac{m^2}{\xi^2} + c_{13}k_z^2\right)J_m(\alpha\xi) - c_{12}\frac{\partial J_m(\alpha\xi)}{\partial\xi} - c_{11}\frac{\partial^2 J_m(\alpha\xi)}{\partial\xi^2}\right]\cos(m\vartheta)e^{-ik_z\eta},$$

$$a_{12z} = 0,$$

$$a_{13z} = 0,$$

$$b_{1z} = -\left(\frac{k_p^2}{\rho\omega^2}\right)\left\langle\frac{1}{1-\left[\frac{r_{01}}{\alpha(a-r_0)}\right]^2}\left\{c_{11}\frac{\partial G_1}{\partial\eta}\frac{\partial^2 J_m\left[\frac{r_{01}}{\alpha(a-r_0)}\xi\right]}{\partial\xi^2} + c_{12}\frac{1}{\xi}\frac{\partial G_1}{\partial\eta}\frac{\partial J_m\left[\frac{r_{01}}{\alpha(a-r_0)}\xi\right]}{\partial r}\right.\right.$$
$$\left.\left. + c_{13}\frac{\partial^3 G_1}{\partial\eta^3}J_0\left[\frac{r_{01}}{\alpha(a-r_0)}\xi\right]\right\}\right\rangle,$$

$$\sigma_{r\theta z} = P_z(a_{21z}A_m + a_{22z}B_m + a_{23z}C_m + b_{2z})e^{-i\omega t}, \tag{A21}$$

where

$$a_{21z} = \frac{(c_{11}-c_{12})}{2}\left[-\frac{ik_z m}{\xi^2}J_m(\alpha\xi) + \frac{ik_z m}{\xi}\frac{\partial J_m(\alpha\xi)}{\partial\xi} - \frac{ik_z m}{\xi^2}J_m(\alpha\xi)\right.$$
$$\left. + \frac{ik_z m}{\xi}\frac{\partial J_m(\alpha\xi)}{\partial\xi}\right]\cos(m\vartheta)e^{-ik_z\eta},$$

$$a_{22z} = 0,$$

$$a_{23z} = 0,$$

$$b_{2z} = 0,$$

and

$$\sigma_{rzz} = \left(a_{31z}A_{mz} + a_{32z}B_{mz} + a_{33z}C_{mj} + b_{3z}\right)e^{-i\omega t}, \tag{A22}$$

where

$$a_{31z} = -c_{44}k_z^2\frac{\partial J_m(\alpha\xi)}{\partial\xi}\cos(m\theta)e^{-ik_z z},$$

$$a_{32z} = 0,$$

$$a_{33z} = 0,$$

$$b_{3z} = -c_{44}\left(\frac{k_p^2}{\rho\omega^2}\right)\left\{\frac{\partial^2 G_1}{\partial\eta^2}\frac{1}{1-\left[\frac{r_{01}}{\alpha(a-r_0)}\right]^2}2\frac{\partial J_m\left[\frac{r_{01}}{\alpha(a-r_0)}\xi\right]}{\partial\xi}\right\}.$$

**Appendix B**

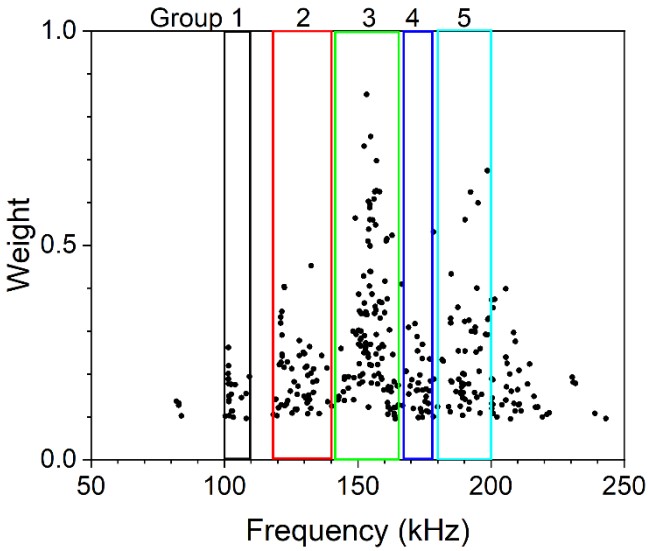

**Figure A1.** Distribution of resolved frequency components and the 5 groups selected using a 10 kHz wide window. The experimental data are obtained from a tensile test using AISI 316 austenitic stainless steel (KS D 3698 standard).

**Table A1.** Resolved frequency components ($\nu$'s) and their relative weights of 316 stainless steel.

| Group | Elements [a] | $\nu$/kHz [b] |
|---|---|---|
| 1 | 20(0.07) | 103.2(2.62) |
| 2 | 58(0.21) | 127.6(5.59) |
| 3 | 114(0.41) | 155.4(5.37) |
| 4 | 36(0.13) | 173.6(3.28) |
| 5 | 51(0.18) | 190.6(4.72) |

[a] Value in parenthesis represents the fraction. [b] Value in parenthesis represents the standard deviation.

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
