# Peer review of "Analytical Modeling of Acoustic Emission Due to an Internal Point Source in a Transversely Isotropic Cylinder"

_applsci, doi:10.3390/app12105272_

Round 1
Reviewer 1 Report
This manuscript derived the displacement fields responsible for acoustic emission excited by a point source in a transversely cylinder. However, I thought it still has some deficiencies and I recommend to revision before acceptable publication. Some comments are listed below:
1: The application background and research meaning of the proposed method should be emphasized in introduction section.
2: Understanding how acoustic waves propagate in media plays an important role in acoustic emission imaging and localization. A more detailed description of previous research in other acoustic emission method field can improve the exposure of the manuscript, such as AE tomography, AE location, and so on. The AE tomography can early detect internal damages, fault and abnormal regions with the distribution of velocity field. Quantitative investigation of tomographic effects in abnormal regions of complex structures. Engineering, 2021, 7(7): 1011-1022. Fracture evolution and localization effect of damage in rock based on wave velocity imaging technology. Journal of Central South University, 2021, 28(9): 2752-2769. The location technique can locate the buried damages and fault with arrivals from active and passive sources. Data-Driven Microseismic Event Localization: An Application to the Oklahoma Arkoma Basin Hydraulic Fracturing Data, IEEE TRANSACTIONS ON GEOSCIENCE AND REMOTE SENSING, 2022(60).
3: The authors needs to further explain the innovation of the proposed method, and what the limitation of previous work is overcomed by the proposed method.
4. The authors need to explain whether the proposed method takes into account the attenuation of acoustic emission waves in the propagation process, and how the faults, joint and other structures in the medium affect the propagation of sound waves.
5: The authors need to further explain how to verify the rationality of the proposed method.
6: The authors needs to explain whether only the single source point is considered, and it is suggested discussing the coupling effect between multiple different types of sources.
Reviewer 2 Report
Paper may have both theoretical and practical significance after thorough revision. Please, find my detailed comments in attached pdf.

Reviewer 3 Report
The paper focuses on the analytical interpretation/modeling of acoustic emission (AE) phenomena in symmetrical solids. In particular, the case study consists in isotropic cylinders, and the structural turbulence/defect is modeled by an internal spatial-temporal concentrate force, also accounting for different source locations. The study provides full formulation development and case study simulation implementations towards a relatively accurate analytical characterization of AE phenomena within structures and components.
The manuscript merits publication after minor amendments are implemented by the Authors. Please, find the review comments within the reviewed manuscript report document.

Round 2
Reviewer 1 Report
Accept in present form
Author Response
The English in this document has been checked by at least two professional editors, both native speakers of English. For a certificate, please see: http://www.textcheck.com/certificate/7VNmG4
Reviewer 2 Report
some unanswered questions:
1) In seismology, the green function is the solution of wave equation with space-time delta function as a source. you solve Poisson equation with impulse source. could you please add an explanation your motivation and implication of results?
2) could you please compare your solution with other published analytical solutions? the physical behaviour of the solution can be better understood considering high-frequency and low-frequency cases.
3) could you please mention results of published numerical studies using finite difference or finite element methods for wave propagation in cylinders. please, explain if there are challenges of these numerical studies in application to acoustic emission monitoring?
4) mentioning results of acoustic emission studies in application to rock deformation would be very useful. Is pre-stress state important to consider in numerical simulation of acoustic emission or not? Please, add a figure with indicated problem setup and boundary conditions.
5) your solution is derived using isotropic elastodynamic equation. I must be stated explicitly without confusing the reader.
Round 3
Reviewer 2 Report
I have found a good explanation and useful references in the response letter but don't see how it has been implemented in the main text. The manuscript file is apparently almost the same as before (if I got a correct version). I recommend that all replies/explanations, additional formulas and references must be implemented in the main text and clearly indicated.
